# YPR-SLAM: A SLAM System Combining Object Detection and Geometric Constraints for Dynamic Scenes

**DOI:** 10.3390/s24206576

**Published:** 2024-10-12

**Authors:** Xukang Kan, Gefei Shi, Xuerong Yang, Xinwei Hu

**Affiliations:** 1School of Aeronautics and Astronautics, Sun Yat-sen University, Shenzhen 518107, China; kanxk@mail2.sysu.edu.cn (X.K.); huxw28@mail2.sysu.edu.cn (X.H.); 2Shenzhen Key Laboratory of Intelligent Microsatellite Constellation (Sun Yat-sen University), Sun Yat-sen University, Shenzhen 518107, China; shigf@mail.sysu.edu.cn

**Keywords:** dynamic scenes, ORB-SLAM2, YOLOv5 network, dense point cloud map, deep learning

## Abstract

Traditional SLAM systems assume a static environment, but moving objects break this ideal assumption. In the real world, moving objects can greatly influence the precision of image matching and camera pose estimation. In order to solve these problems, the YPR-SLAM system is proposed. First of all, the system includes a lightweight YOLOv5 detection network for detecting both dynamic and static objects, which provides pre-dynamic object information to the SLAM system. Secondly, utilizing the prior information of dynamic targets and the depth image, a method of geometric constraint for removing motion feature points from the depth image is proposed. The Depth-PROSAC algorithm is used to differentiate the dynamic and static feature points so that dynamic feature points can be removed. At last, the dense cloud map is constructed by the static feature points. The YPR-SLAM system is an efficient combination of object detection and geometry constraint in a tightly coupled way, eliminating motion feature points and minimizing their adverse effects on SLAM systems. The performance of the YPR-SLAM was assessed on the public TUM RGB-D dataset, and it was found that YPR-SLAM was suitable for dynamic situations.

## 1. Introduction

Simultaneous localization and mapping (SLAM) is one of the most important topics in these fields [1,2,3]. SLAM is a technique that makes use of onboard sensors to estimate one’s location and to build an environmental map at the same time. It is widely used in intelligent vehicles and augmented reality, among other fields [4,5,6]. SLAM can be classified into Lidar SLAM and Visual SLAM, depending on the type of sensor used [7,8]. Lidar SLAM acquires depth information of object surfaces through Lidar. Lidar emits Lidar beams and calculates the distance to the target by measuring the reflection time. While moving, the system can construct a 3D image of the environment and also estimate the movement status of the mobile platform through multiple measurements. Visual SLAM is based on a visual camera as a sensor to estimate the movement of the camera. Compared to Lidar SLAM, Visual SLAM has the advantages of lower cost, the ability to provide richer semantic information, and better suitability for complex environments. Therefore, visual SLAM is getting more and more attention and is a hot topic of study [9,10].

As SLAM continues to evolve, it has been widely applied in various platforms, such as monocular, stereo, and RGB-D systems. Many outstanding SLAM systems have been proposed. For example, in 2007, Davison et al. introduced Mono-SLAM, the first real-time SLAM system [11]. Subsequently, Klein et al. proposed a PTAM system [12] that allows accurate estimation of camera posture in an unfamiliar environment. In 2015, Mur-Artal et al. improved the PTAM system and proposed the ORB-SLAM system [13], which innovatively introduced a three-threaded SLAM system. In 2017, they further released ORB-SLAM2 [14], which supports monocular, stereo, and RGB-D modes. Building on this, they developed ORB-SLAM3 [15], which incorporates inertial measurement units for multi-map construction. Despite the significant achievements of current Visual SLAM frameworks in static environments, many issues persist in dynamic environments. For example, the accuracy and robustness of map updates and pose estimation are significantly reduced when dynamic objects are present in the scene [16,17,18]. Therefore, it is very important to improve the localization precision of the SLAM system in a dynamic environment.

At present, there are two main approaches to SLAM in a dynamic environment: the geometric approach and the semantic approach [19,20]. Geometric methods detect and exclude outliers or dynamic regions with large residuals using multi-view geometry or other traditional techniques. In contrast, deep learning techniques, such as object detection and instance segmentation, can be used to detect and segment objects [21]. Many scholars have conducted in-depth research on SLAM in dynamic environments and proposed various solutions. At present, SLAM frameworks for dynamic environments include Dyna-SLAM [22], DS-SLAM [23], and Dynamic-SLAM [24]. Dyna-SLAM introduces the Mask-RCNN network [25] for semantic segmentation, then uses multi-view geometry methods to detect outliers, generates dynamic detection regions through region growing, and filters out the dynamic regions identified by multi-view geometry with those segmented by semantics. Finally, the masked image is fed into the tracking thread. Zhang et al. proposed the PR-SLAM system [26], a real-time SLAM system with several improvements based on ORB-SLAM3. Firstly, a semantic optimization thread decoupled from the tracking thread was added, employing a semantic segmentation model with a dynamic probability update strategy to improve accuracy. Secondly, a new method based on reprojection error was proposed to compensate for the semantic gaps generated by the semantic segmentation model. In order to ensure real time performance and reduce semantic gaps, SOLOv2 [27] has been lightly modified.

He et al. proposed an ORB-SLAM3 on-line SLAM system [28]. The system uses the combination of semantic and light flow to filter out the foreground points in those areas as dynamic points. Finally, the static point on the moving target is recovered by means of the mean re-projection error, which is suitable for the non-rigid motion and the low-dynamic environment. Cong et al. introduced the SEG-SLAM system [29] for dynamic scenes. Based on ORB-SLAM3, YOLOv5 is used to construct a new model of object detection. Based on the prior information, the depth information, and the geometric approach, the SLAM system localization and the construction of the map can be improved. At last, we combine the filtered data with the depth information to build the 3D map of the static point cloud with the point cloud database.

Islam QU et al. proposed an ARD-SLAM system [30], which is a combination of global dense optical tracking and complicated geometry. The key innovation of the system is its dynamic object recognition, which combines the geometry information with the desired motion data, which greatly reduces the influence of the motion estimation on the camera. Li et al. introduced the DDN-SLAM system [31], a dense dynamic neural implicit SLAM system that integrates semantic features. Semantic features are first combined with a mixture of Gaussian distribution models for feature point segmentation, and then, based on sparse point cloud sampling and background recovery mapping strategy, a novel semantic loss is proposed to eliminate dynamic occlusion.

The aforementioned methods have made progress in addressing the issue of decreased SLAM localization accuracy in dynamic scenes, but several significant challenges remain. For instance, systems like DynaSLAM rely on the Mask-RCNN deep learning network, which requires inference and segmentation of the entire image. This significantly increases the system’s computational load and resource demands, placing pressure on the system and affecting its real-time performance. Furthermore, some systems can mistakenly classify static feature points as dynamic, resulting in a shortage of valid static feature points. This reduction in reliable static feature points directly impacts the accuracy of localization by limiting the information available for pose estimation. Additionally, during the dense mapping process, these systems may incorrectly integrate dynamic objects into the map, leading to severe map inconsistencies and artifacts, such as ghosting, which undermine the map’s overall quality and usability. This highlights the need for more robust handling of dynamic environments to improve both real-time performance and map integrity.

In order to solve these problems, YPR-SLAM is proposed. This system first introduces a YOLOv5 [32,33] network, which can detect potentially dynamic objects. And it provides reliable advance information about dynamic targets. Secondly, the method combines the depth map with the geometry method to recognize and filter the dynamic feature point. At last, a dense 3D point cloud map is constructed on the basis of the static feature points. The main contributions of this paper are as follows:(1)In order to solve the problem of reduced SLAM location precision caused by dynamic objects, it is proposed to integrate the YOLOv5 algorithm of the potential moving target detection module into the ORB-SLAM2 system. This module meets real-time requirements and effectively extracts prior information about moving objects, thereby enhancing the system’s adaptability to dynamic environments.(2)Innovatively tackling the complexity of dynamic feature point filtering, this paper introduces the Depth-PROSAC algorithm. The Depth-PROSAC algorithm makes use of the pixel coordinates in the bounding box provided by the motion object detection module to precisely filter out the real dynamic feature points in the detection box.(3)Addressing issues such as “ghosting” in dense point cloud map construction under dynamic environments, this article proposes a method to construct dense 3D point cloud maps using retained static feature points. This approach effectively reduces interference from dynamic objects in map construction, thereby enhancing map reliability. The proposed SLAM system is assessed by the TUM RGB-D dataset [34]. Experiments show that the system is effective and superior in a dynamic environment.

The rest of the paper is structured as follows: Section 2 introduces an overview of the YPR-SLAM System. Section 3 evaluates the algorithm of this paper in the TUM datasets. Section 4 is the discussion and conclusion of this paper.

## 2. Overview of YPR-SLAM System

The YPR-SLAM system that is designed for dynamic environments is presented in this section. First of all, a lightweight object detection algorithm, YOLOv5, is integrated into the ORB-SLAM2 TRACKING thread. The YOLOv5 algorithm can efficiently detect potential dynamic objects, ensuring real-time performance while providing a good prior region. At present, the usual way to deal with the problem of a dynamic environment is to remove the feature points that are extracted from the area of the detected object [35,36,37]. However, when the detected region occupies two-thirds or more of the image, the method’s effectiveness decreases [19,38]. Additionally, if the region contains a large number of high-quality feature points, the method’s performance is further compromised. To solve this issue, this paper proposes the Depth-PROSAC method. Based on the depth information, the feature points in the detection box can be classified as dynamic or static. This makes it possible to accurately identify and filter the dynamic feature points. As a result, the precision and robustness of the SLAM system are further improved. In addition, this paper provides the ability of dense 3D point cloud mapping. A general framework will be presented, which will detail the functions and functions of the various components of a YPR-SLAM system.

### 2.1. YPR-SLAM Framework

Figure 1 illustrates the framework of YPR-SLAM. The system is composed of five threads: TRACKING, Dynamic Target Detecting and Filtering, LOCAL MAPPING, LOOP CLOSURE, and Dense 3D point cloud mapping. These five threads operate independently but collaborate to perceive the environment and perform mapping tasks, ensuring the system’s efficiency and accuracy.

### 2.2. Lightweight Dynamic Target Detection Thread

It is a complicated task for SLAM in dynamic environments. A lot of research has introduced the method of semantic segmentation that is used to detect the latent dynamic object. The high computational demands of semantic segmentation models often exceed the processing capabilities of embedded devices, preventing the system from responding in real time. Therefore, reducing the computational complexity while maintaining system accuracy has become a critical issue.

In order to solve this problem, a lightweight object detection algorithm named YOLOv5 is presented. The network structure is illustrated in Figure 2. YOLOv5 is a classical single-step object detection algorithm that has been widely used in recent years. First, in terms of accuracy, YOLOv5 achieves an average precision of 50.5% on the COCO dataset, while YOLOv8 achieves 51.4%. Second, in terms of speed (FPS), YOLOv5 is highly optimized, reaching 98 ms in the CPU version, whereas YOLOv8 takes 128.4 ms. Regarding parameters, YOLOv5s has 7.2 M parameters, while YOLOv8s has 11.2 M. Although YOLOv8’s increased number of parameters slightly improves detection accuracy, it sacrifices detection speed. Since edge devices need to consider memory and the runtime speed of SLAM algorithms, YOLOv5 is chosen [39]. By optimizing network architecture and using efficient convolution operations, it achieves real-time detection on embedded devices. Compared to traditional deep learning segmentation models, YOLOv5 significantly reduces computational complexity, ensuring detection accuracy while meeting the resource constraints of embedded devices. The specific process is as follows:

Firstly, the input color image is preprocessed, including scaling and normalization operations. After the preprocessing module, the image enters the BackBone module. This module uses the CSP-Darknet53 neural network, which extracts image features through multiple convolutional and pooling layers, forming high-level feature representations.

Next, the features enter the Neck module. The Neck module uses FPN and PAN structures to fuse features at different scales. Finally, the features enter the Prediction module, generating multiple prediction boxes. Each prediction box contains a confidence score and class information. Additionally, each prediction box is classified to determine its category. The non-maximum suppression (NMS) method is used to remove overlapping boxes and retain the optimal detection results.

To improve detection accuracy, YOLOv5 uses the *CIoU* loss function, which is calculated as follows:(1)$${\mathrm{CIoU}(B,B}_{gt})=\mathrm{IoU}(B,B_{gt})-\frac{\rho^{2}(B,B_{\mathrm{gt}})}{c^{2}}-\alpha v$$
(2)$$v=\frac{4}{\Pi }{\left(\arctan\frac{w^{gt}}{h^{gt}}-\arctan\frac{w}{h}\right)}^{2}$$
(3)$$\alpha =\frac{v}{1-\mathrm{IoU}(B,B_{gt})+v}$$
where v is the normalized difference in aspect ratio between the predicted box and the ground truth box. ${\left(\arctan\frac{w^{gt}}{h^{gt}}-\arctan\frac{w}{h}\right)}^{2}$ range between 0 and $\frac{\Pi }{4}$. α is a balancing factor to balance the loss due to the length-width ratio and the loss due to the *IoU* component.

The introduction of YOLOv5 enables effective detection of potential dynamic objects, providing prior dynamic object regions for subsequent feature point filtering. This algorithm completes object detection in a single forward pass, making it highly efficient. Its core idea is to directly predict bounding boxes and class probabilities on the input image without complex multi-stage processing. This not only improves detection speed but also reduces computational resource consumption.

The process of the algorithm is shown in Figure 3. In the YPR-SLAM system, the YOLOv5 network is introduced to obtain the pre-information of dynamic and static objects. Firstly, a latent dynamic target detecting module processes the input RGB image for every frame. The YOLOv5 algorithm, with its efficient convolutional neural network, detects dynamic and static objects in the image, generating 2D detection results that include object bounding boxes and category labels. These detection box details consist of the bounding box coordinates (top-left and bottom-right corners) and object category labels (e.g., “person,” “car”). Each frame of the image is detected by the YOLOv5 algorithm, which generates the coordinate points of the bounding boxes for each target in the image coordinate system, as well as the label information. The coordinates of the 2D bounding boxes for each detected target and the target’s category (such as “person” or “car”) are used as inputs to the YPR-SLAM system. Through the integration of the lightweight YOLOv5 detection algorithm and the ORB-SLAM2 system, the YPR-SLAM system is able to realize efficient and accurate perception and mapping in dynamic environments.

### 2.3. Dynamic Feature Point Filtering Algorithm

SLAM systems rely heavily on feature points for overall stability and performance. Dynamic feature points may result in errors in route estimation. To overcome this problem, some researchers have adopted the method of removing all the feature points from the bounding box. Although the method is effective in reducing the dynamic disturbance, it also removes many of the static feature points. The loss of static feature points weakens the feature matching constraints and therefore reduces the resolution accuracy. This problem is especially pronounced when the detection box takes up a large part of the current frame, which may lead to significant errors.

In this system, the Depth-PROSAC algorithm is proposed to tackle these challenges by leveraging depth information from depth images. The process of the algorithm is shown in Algorithm 1. The Depth-PROSAC algorithm distinguishes feature points within potential target detection boxes based on depth information. Since dynamic objects exhibit different depth behaviors in depth images, with their depth values fluctuating within a small range, the Depth-PROSAC algorithm effectively identifies and filters out these dynamic feature points. The algorithm proceeds as follows: it combines depth images with the PROSAC method to differentiate and filter feature points within potential target detection boxes. Lastly, it filters out feature points that lie inside dynamic object boxes but not within static object boxes, checking their depth validity and storing them in a vector container. These feature points are then subjected to iterative filtering: in each iteration, two feature points are selected at random, their depth difference is calculated, and the inlier count is determined on the basis of the depth difference. At last, the algorithm chooses the optimal model with the largest number of inliers and makes the corresponding feature points as dynamic feature points. As illustrated in Figure 4, this method is effective in filtering dynamic feature points, thus increasing the stability of SLAM.

**Algorithm 1.** Dynamic feature point filtering algorithm. **Technique:** Depth-PROSAC algorithm**Input:** YOLOv5 detects bounding boxes of dynamic objects, box_i; The extracted ORB feature points, mvkey; Depth threshold, nThre;The depth image, DImg**Output:** The set of ORB feature points excluding dynamic feature points, mvkey1. Identify feature points inside potential dynamic object detection boxes that are not inside static detection boxes:mvKeysDynamic = FindPoints_Within_dynamicBBox_without_staticBBox(box_i, mvkey)2. Sort mvKeysDynamic based on priority in descending order.3. Initialize the maximum number of inliers and iteration count:nbestInliers = 0nIter = 04. For each iteration (n < k):Set the number of inliers to zero and initialize a temporary set for dynamic points:nInliers = 0tempDynamicPoints = []5. Define the progressive sample set size:sampleSize = min(length(mvKeysDynamic), nIter + 2)6. Randomly select any two points from the first sampleSize points in mvKeysDynamic:var1 = CalcInitialDepthVariance(Dpoint1, Dpoint2)7. For each of the remaining points in the first sampleSize points in mvKeysDynamic:Add the point and calculate:var2 = CalcDepthVariance(var1, Dpoint3)8. If var2 < nThre, then:Increment the number of inliers:++nInliersAppend Dpoint3 to tempDynamicPoints9. If nInliers > nbestInliers:Update the best inliers count:nbestInliers = nInliers10. Increment the iteration count:++nIter11. End for12. Return mvkey excluding the dynamic feature points.

### 2.4. Static Dense 3D Point Cloud Map

Constructing maps is a core task in visual SLAM. Sparse point cloud maps built in dynamic environments cannot be directly used for higher-level applications such as navigation. Additionally, the presence of dynamic objects when constructing dense point cloud maps can lead to issues such as “ghosting.” In this paper, a new approach is presented to construct dense cloud maps with static feature points.

Using RGB-D camera and depth map transformation, a 3D point cloud map is produced, and an extra thread is added to the ORB-SLAM2. In this thread, the RGB-D camera is used to match the RGB image, and the space coordinates of all the pixels are calculated according to the position of the camera, and the point cloud map is generated. Assume that the coordinates of a 3D point X in space are $X{=[X}_{w},Y_{w},Z_{w}]$. The pixel coordinates corresponding to a 3D spatial point in a camera coordinate system are x=[μ,v,1]. According to the pinhole camera imaging model, it can be obtained that:(4)$$z\left[\begin{array}{l} u\\ v\\ 1 \end{array}\right]=K(R+t)\left[\begin{array}{l} X_{w}\\ Y_{w}\\ Z_{w}\\ 1 \end{array}\right]$$

Since the camera has been calibrated, the intrinsic parameter matrix K is known, and TRACKING calculates the pose, thus determining R and t. Therefore, the spatial coordinates of each pixel in the ORB image can be determined. Using Equation (4), the 3D point cloud coordinates can be obtained:(5)$$\left\{\begin{array}{l} X_{w}=\frac{z(u-c_{x})}{f_{x}}\\ Y_{w}=\frac{z(u-c_{y})}{f_{y}}\\ Z_{w}=z \end{array}\right.$$

The workflow of the dense 3D point cloud mapping thread consists of three parts: data input, pose estimation, and map construction, as shown in Figure 5. Firstly, the necessary data for mapping, including RGB images and depth maps, is input. Then, the ORB-SLAM2 system is used to provide pose estimation for keyframes that meet the mapping requirements. At last, we get the image data and pose information and finish the 3D point cloud stitching.

## 3. Simulation and Discussion

In this section, we will give a comprehensive evaluation of YPR-SLAM’s performance and show it from four points of view. First of all, the datasets used in the experiments and the evaluation measures are presented. Secondly, based on the TUM RGB-D dataset, the proposed algorithm is verified. Through this comparison, there can be a more intuitive display of the system in the face of a dynamic environment with the advantages. Thirdly, this paper compares the proposed system with the existing SLAM system, and evaluates the performance differences in various application environments. In the end, we summarize the advantages and disadvantages of the algorithm in both precision and real-time. The simulation experiments were conducted on one desktop computer equipped with an Dell workstation(Intel^®^ Xeon^®^ Platinum 8160 CPU, 2.10 GHz clock speed, 128 GB RAM, NVIDIA Quadro P4000 graphics card, and Ubuntu 18.04 operating system). The comparison algorithm was run on a MECHREVO laptop (a 12th Gen Intel^®^ Core™ i5-12450H CPU, 2.00 GHz clock speed, 16.0 GB RAM, NVIDIA GeForce RTX 3060 graphics card, and Ubuntu 20.04 operating system). For the prior dynamic object module, the input size of the RGB-D images was 640 × 640. In all experiments, the feature point number N was set to 1000, and the depth threshold d_t_ was set to 0.3 m.

### 3.1. Experimental Conditions

In this paper, the TUM RGB-D dataset is widely considered as a benchmark for evaluating the performance of SLAM systems. The TUM dataset was captured with a Kinect camera. It consists of an RGB-D image, a depth image, and ground track data. The dataset consists of multiple sequences such as sitting, walking, and desk scenes. In view of the fact that SLAM is a dynamic environment, experiments are carried out on the following four sequences: fre3_walking_xyz, fre3_walking_halfsphere, fre3_walking_static, and fre3_sitting_static. The walking sequences cover a variety of scenarios, including high and low dynamic scenes. In high-motion scenes, the two of them are walking around a table, while the low-motion scenes depict two people sitting inside, making gestures and communicating verbally. These sequences feature diverse camera motion trajectories, including movements along the x, y, and z axes while maintaining direction, stationary positions, rotations along principal axes (roll-pitch-yaw), and movement along a one-meter diameter half-sphere.

Typically, when evaluating map construction accuracy, direct observation of map construction results provides an intuitive understanding of its effectiveness, albeit this observation only describes the relative accuracy of an algorithm in map construction. So, in order to evaluate the precision of the map, we select absolute trajectory error (*ATE*) and relative pose error (*RPE*) as the error measure [40].

*ATE* is the absolute deviation between the measured trajectory and the actual trajectory. This index directly reflects the overall accuracy of the method and the consistency of global tracking. Through the analysis of *ATE*, the accuracy and stability of the algorithm can be evaluated. The difference between any estimated frame trajectory and the ground truth trajectory is calculated as:(6)$$E_{i}=P_{\mathrm{ref},i}^{-1}P_{\mathrm{est},i}\in \mathrm{SE}(3)$$
where $P_{\mathrm{ref},i}$ represents the estimated camera pose and $P_{\mathrm{est},i}$ represents the true camera trajectory.

Therefore, according to Equation (6), the formula for *ATE* calculation is:(7)$$\mathrm{ATE}=\frac{1}{N}\sum_{i=1}^{N}\left\|\mathrm{tran}s(E_{i})\right\|$$

The calculation formula for *RMSE* in *ATE* is as follows:(8)$$\mathrm{RMSE}=\sqrt{\frac{1}{N}{\sum_{i=1}^{N}\left\|\mathrm{trans}(E_{i})\right\|}^{2}}$$

*RPE* describes the difference between estimated and true trajectories of adjacent frames, serving to assess the robustness of SLAM systems under consecutive pose estimations. *RPE* primarily focuses on the relative error between continuous pose estimates. In this experiment, *RMSE* and *S.D.* are chosen as quantitative metrics to evaluate algorithm accuracy.
(9)$$E_{i,j}={\left(P_{ref,i}^{-1}P_{ref,j}\right)}^{-1}(P_{est,i}^{-1}P_{est,j})\in \mathrm{SE}(3)$$
where $P_{\mathrm{ref},i}$ represents the estimated camera pose of the camera at frame *i*, and $P_{\mathrm{est},i}$ represents the real camera trajectory at frame *i*.

*RPE* is calculated as:(10)$$RPE=\frac{1}{N}\sum_{i=1}^{N}\left\|\mathrm{tran}s(E_{i,j})\right\|$$

The *RMSE* for *RPE* is calculated as follows:(11)$$\mathrm{RMSE}=\sqrt{\frac{1}{N}{\sum_{\forall \sim i,j}^{N}\left\|\mathrm{trans}(E_{i,j})\right\|}^{2}}$$

The following formula is used to calculate the improvement in performance:(12)$$\eta =\frac{\eta_{ORB-SLAM2}-\eta_{ours}}{\eta_{ORB-SLAM2}}\times 100\%$$

### 3.2. Positioning Accuracy Analysis

Figure 6 shows the comparison of the camera trajectories estimated by the ORB-SLAM2 and the YPR-SLAM systems in the high dynamic scene of the sequence fr3_walking_halfsphere. It is evident from the figure that the YPR-SLAM system achieves more precise camera trajectory estimation, with better alignment to the groundtruth trajectory compared to the ORB-SLAM2 system. Specifically, YPR-SLAM demonstrates higher robustness and stability in dynamic environments. The improved method not only improves the precision of map construction but also enhances the reliability of the system in practice. The ORB-SLAM2 system, on the other hand, shows a certain degree of bias and error when estimating trajectory in the same environment. Through comparative analysis, it is clear that the YPR-SLAM system shows superior performance in dynamic environments.

Figure 7 shows the absolute trajectory error (*ATE*) and relative pose error (*RPE*) calculated using ORB-SLAM2 and YPR-SLAM. For the low dynamic scene sequence fre3_siting_static, the errors of ORB-SLAM2 and YPR-SLAM are very close to the real trajectory on the ground, indicating that both systems can provide relatively accurate trajectory estimation in this scene. However, in another low dynamic scene sequence, fre3_walking_static, the absolute trajectory error and relative pose error of ORB-SLAM2 drift significantly, showing that the system lacks robustness in this scenario. In contrast, the YPR-SLAM system maintains high accuracy, and its trajectory estimation remains stable in this scene. Further analysis of the high dynamic scene sequences fre3_walking_halfsphere and fre3_walking_xyz reveals that the trajectory estimation of ORB-SLAM2 deviates significantly from the real trajectory, with a considerable increase in error. This demonstrates the limitations of ORB-SLAM2 in handling high dynamic environments. Conversely, the absolute trajectory error of the YPR-SLAM system in these high dynamic scenes is smaller and closer to the real trajectory, showing its excellent performance in such environments. From these experimental results, it is evident that the YPR-SLAM system performs superiorly in dynamic environments, especially in high dynamic scenes where it exhibits better robustness and accuracy. In contrast, ORB-SLAM2’s performance in dynamic scenes is relatively weaker, as it is more susceptible to environmental changes, leading to increased errors. The YPR-SLAM system demonstrates higher precision and consistency under different dynamic conditions, with its advantages being even more pronounced in high dynamic scenes, proving its reliability in complex dynamic environments. These results further verify the applicability of the YPR-SLAM system in dynamic environments, providing more stable and accurate trajectory estimation for real-world applications.

Table 1 presents the impact of using the YOLOv5 detection algorithm and the Depth-PROSAC algorithm, either separately or together, on the system’s performance, with the *RMSE* of absolute trajectory error as the evaluation metric and ORB-SLAM2 as the baseline. YPR-SLAM (YOLO) refers to the system that solely uses the YOLOv5 algorithm to detect dynamic objects and remove dynamic feature points within the detection bounding boxes. YPR-SLAM (Depth-PROSAC), on the other hand, utilizes the depth map and PROSAC algorithm to process the entire image, classifying dynamic and static feature points without specifically targeting dynamic objects. YPR-SLAM (YOLO + Depth-PROSAC) represents the final system proposed in this paper, which combines both YOLOv5 and Depth-PROSAC algorithms. In YPR-SLAM, potential dynamic object regions are first detected, and then the feature points within these regions are further classified, aiming to retain as many static feature points as possible while eliminating dynamic ones. As shown in Table 1, YPR-SLAM achieves the highest accuracy through this approach, effectively optimizing the removal of dynamic feature points and the preservation of static ones.

Table 2 compares and analyzes the results of the *RMSE* and the *S.D.* of YPR-SLAM, ORB-SLAM2, and DS-SLAM. These comparisons give a quantitative insight into how much the YPR-SLAM system has improved over the others.

As shown in Table 2, *RMSE* of the YPR-SLAM system is increased by 94.78% compared with the ORB-SLAM2 system, and *S.D.* is increased by 94.17% in comparison with the ORB-SLAM2. In the fre3_walking_halfsphere sequence, *RMSE* of the YPR-SLAM system can be improved by 93.85% and *S.D.* by 92.45% compared with ORB-SLAM2. In the fre3_walking_static sequence, although the improvement of the YPR-SLAM system is relatively smaller, it still significantly outperforms the ORB-SLAM2 system, with *RMSE* increased by 69.38% and *S.D.* increased by 71.17%. The improvement of the YPR-SLAM algorithm to sequence fr3_sitting_static is not obvious to other high dynamic sequences. The reason is that the traditional SLAM technology has already possessed the ability of positioning in static and low dynamic environments. However, even in these scenes, the YPR-SLAM system shows moderate performance improvement, with *RMSE* increased by 14.93% and *S.D.* increased by 18.75% compared to the ORB-SLAM2 system.

Comparing with the DS-SLAM system, it can be seen that the YPR-SLAM system performs better on all experimental datasets and exhibits lower absolute trajectory error. This is because the YPR-SLAM system first detects potential dynamic regions in dynamic scenes, filters out true dynamic feature points using the Depth-PROSAC algorithm, and preserves more static feature point information as much as possible, thereby achieving better results in subsequent feature matching and pose estimation.

In Table 3 and Table 4, the relative posture errors are represented, respectively, by the relative translation vector errors and the relative rotational matrix errors. From Table 3, we can see that the YPR-SLAM system excels at high dynamic scenes with respect to relative translation vector errors. In fre3_walking_xyz sequence, *RMSE* was increased by 92.10%, and *S.D.* was 94.42% higher than that of ORB-SLAM2. In the fre3_walking_halfsphere sequence, *RMSE* increases by 88.35% and *S.D.* increases by 91.90%. In the low dynamic scene fre3_walking_static, the YPR-SLAM system shows *RMSE* increases of 94.31% and *S.D.* increases by 96.77%. Even in the static scene fr3_sitting_static, the YPR-SLAM system demonstrates improvement, with *RMSE* increasing by 15.79% and *S.D.* by 26.79%.

Table 3 displays relative rotation matrix errors, showing that the YPR-SLAM system continues to perform exceptionally well in high dynamic scenes. In the fre3_walking_xyz sequence, *RMSE* increases by 87.66% and *S.D.* increases by 88.85% compared to the ORB-SLAM2 system. In the fre3_walking_halfsphere sequence, RMSE increases by 84.63% and *S.D.* increases by 89.31%. In the fre3_walking_static sequence, *RMSE* increases by 92.40%, and *S.D.* increases by 96.13%. In the static scene fr3_sitting_static, the YPR-SLAM system shows *RMSE* increases of 3.98% and *S.D.* increases by 9.37%.

Overall, the YPR-SLAM system has demonstrated remarkable performance advantages over the ORB-SLAM2 system, not only in absolute trajectory errors but also in relative translation and rotation errors. These results show that the YPR-SLAM system is robust and stable in dealing with dynamic environments.

### 3.3. Real-Time Analysis

Real-time performance is a crucial metric in practical applications. Table 5 compares the runtime performance of various SLAM systems across different sequences, highlighting the effectiveness of the YPR-SLAM system in relation to other systems. The evaluation is based on the median and mean processing times per frame for each system. The best values are highlighted in bold to emphasize superior performance. This table clearly demonstrates the superior performance of the YPR-SLAM system, achieving significantly faster processing times compared to ORB-SLAM2 and DS-SLAM across all tested sequences.

From Table 5, we can see that the total computing time of the YPR-SLAM system for the fre3_walking_xyz sequence is much less than that of ORB-SLAM2 and DS-SLAM. Specifically, the median time for the YPR-SLAM system is 37.74 milliseconds, and the mean time is 38.94 milliseconds, whereas for the ORB-SLAM2 system, the median and mean times are 50.93 milliseconds and 51.90 milliseconds, respectively, and for the DS-SLAM system, they are 57.98 milliseconds and 59.12 milliseconds.

While maintaining high robustness, the YPR-SLAM system also demonstrates excellent overall operational efficiency. This advantage is attributed to its use of a lightweight single-stage object detection approach to handle potential dynamic objects, ensuring good real-time performance. In addition, the YPR-SLAM can effectively remove the real dynamic feature points by using the Depth-PROSAC method, which further improves its real time performance and robustness.

These results indicate that the YPR-SLAM system not only outperforms other systems in terms of accuracy and robustness but also exhibits significant advantages in real-time performance and efficiency, making it an ideal choice for practical applications.

### 3.4. Dense Point Cloud Mapping

Figure 8 illustrates the performance of the ORB-SLAM2 and YPR-SLAM systems in constructing the fre3_walking_xyz sequence, a highly dynamic scene. From Figure 8, it can be seen that the dense 3D point cloud map produced by ORB-SLAM2 includes dynamic objects, which results in bad consistency of map and “ghost”. In contrast, the YPR-SLAM system eliminates feature points associated with dynamic objects, retaining only static feature points. As a result, it can construct a denser 3D point cloud map with better consistency.

## 4. Conclusions

Aiming at the low localization accuracy of the SLAM system in a dynamic environment, YPR-SLAM is presented. Firstly, the system uses the YOLOv5 algorithm to identify the previous area of the potential dynamic target. Then, the Depth-PROSAC algorithm is used to get rid of the identified dynamic feature points. Then, the remaining static feature points are used to construct a dense 3D point cloud map. Experimental results indicate that the reduction in the ATE in the YPR-SLAM system is 94.78% compared with ORB-SLAM2. This work has broader implications for various real-world applications. The enhanced localization accuracy opens up possibilities for mobile robots in areas such as autonomous navigation, dynamic object tracking, and smart infrastructure in complex environments like urban spaces, factories, and warehouses. Additionally, this work contributes to the development of more robust SLAM systems capable of adapting to changing environments, which is critical for future advancements in autonomous systems, including autonomous vehicles and drones.

As for future work, there are several promising directions to explore. One potential area is to further optimize the computational efficiency of the system, making it more suitable for real-time applications on lower-power devices. Another direction could involve integrating additional sensor modalities such as LiDAR or radar to improve accuracy in environments with limited visual information. Furthermore, extending the system to handle semi-dynamic environments where some objects exhibit periodic or intermittent motion could further enhance its robustness in real-world scenarios.

## Figures and Tables

**Figure 1 sensors-24-06576-f001:**
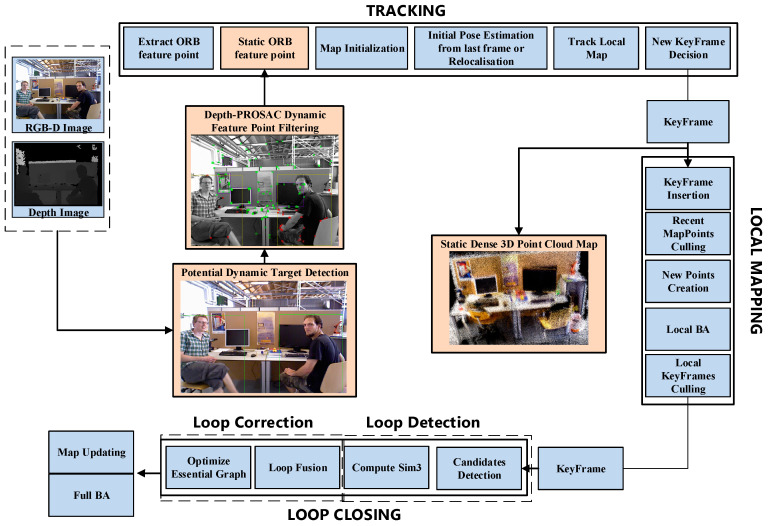
Framework of YPR-SLAM System. The blue section is ORB-SLAM2, and the orange section is the addition of this paper.

**Figure 2 sensors-24-06576-f002:**
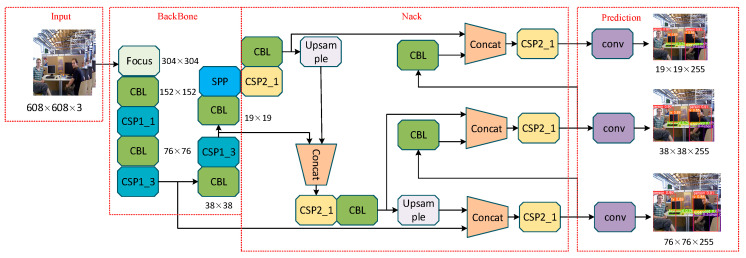
The YOLOv5 network architecture.

**Figure 3 sensors-24-06576-f003:**
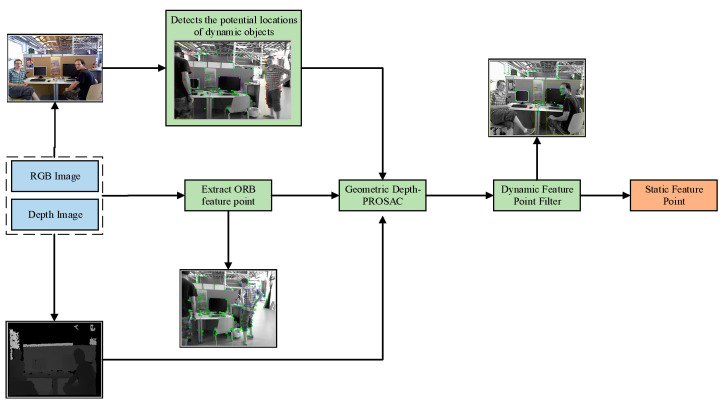
Dynamic target detection and filtering thread. First, the ORB feature point is extracted from the RGB image by the tracking thread. Next, the dynamic target detection thread identifies potential dynamic target areas, and then the Depth-PROSAC algorithm is applied to filter out dynamic feature points. Finally, the static feature points are retained for subsequent pose estimation.

**Figure 4 sensors-24-06576-f004:**
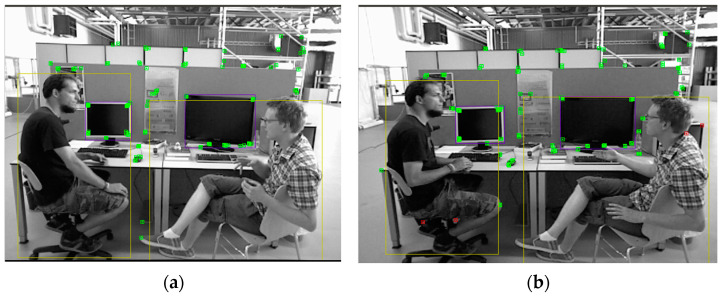
The comparison between target detection algorithms and the Depth-PROSAC algorithm in filtering out dynamic feature points. (**a**) shows that the object detection method directly filters out dynamic feature points, and (**b**) shows that the Depth-PROSAC algorithm filters out dynamic feature points.

**Figure 5 sensors-24-06576-f005:**
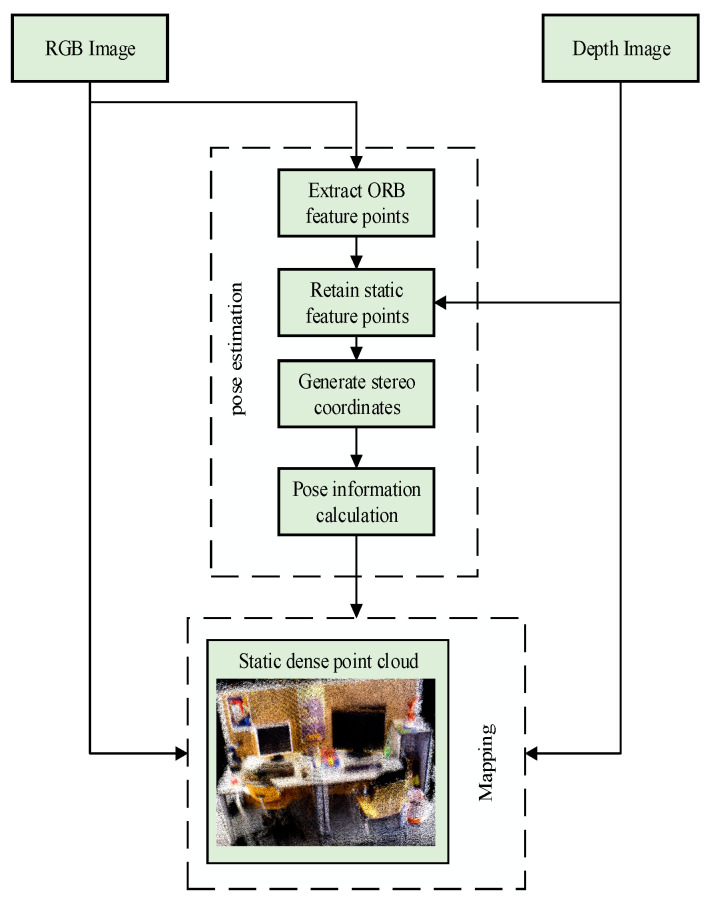
Dense point cloud construction workflow.

**Figure 6 sensors-24-06576-f006:**
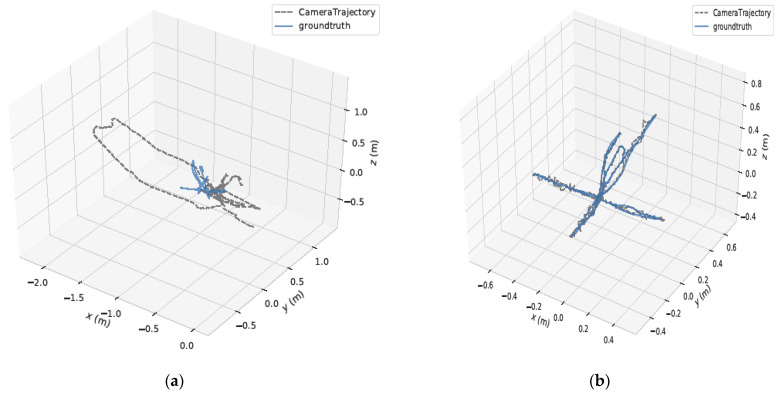
In the fr3_walking_halfsphere sequence, the YPR-SLAM and ORB-SLAM2 systems were used to estimate the 3D motion of the camera. (**a**) Camera path estimated by ORB-SLAM2; (**b**) YPR-SLAM estimation of camera trajectory.

**Figure 7 sensors-24-06576-f007:**
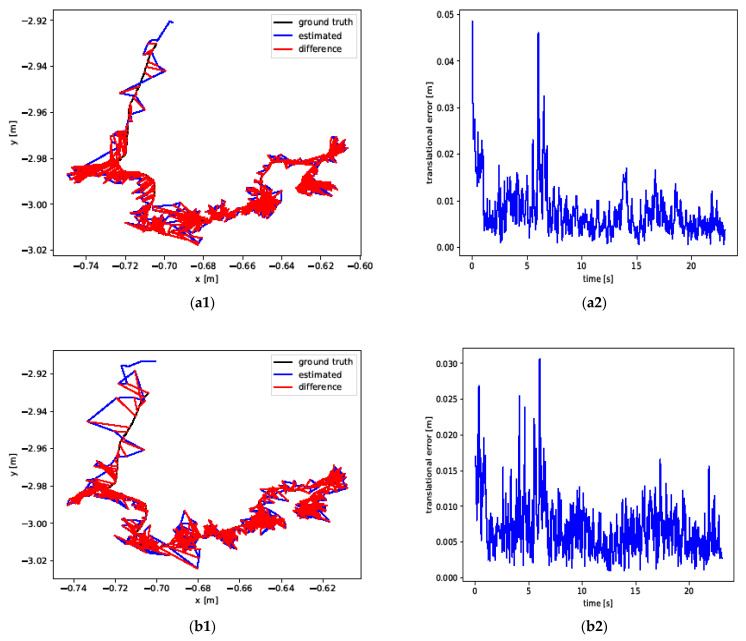
*ATE* and *RPE* of the ORB-SLAM2 system and the YPR-SLAM system under different datasets. (**a1**,**a2**,**c1**,**c2**,**e1**,**e2**,**g1**,**g2**) represent ATE and RPE obtained by the ORB-SLAM2 system by running fre3_sitting_static, fre3_walking_static, fre3_walking_halfsphere, and fre3_walking_xyz, respectively. (**b1**,**b2**,**d1**,**d2**,**f1**,**f2**,**h1**,**h2**) represent *ATE* and *RPE* plots of the YPR-SLAM system running fre3_sitting_static, fre3_walking_static, fre3_walking_halfsphere, and fre3_walking_xyz, respectively. (**a1**,**b1**,**c1**,**d1**,**e1**,**f1**,**g1**,**h1**) represent ATE plots. (**a2**,**b2**,**c2**,**d2**,**e2**,**f2**,**g2**,**h2**) represent *RPE* plots.

**Figure 8 sensors-24-06576-f008:**
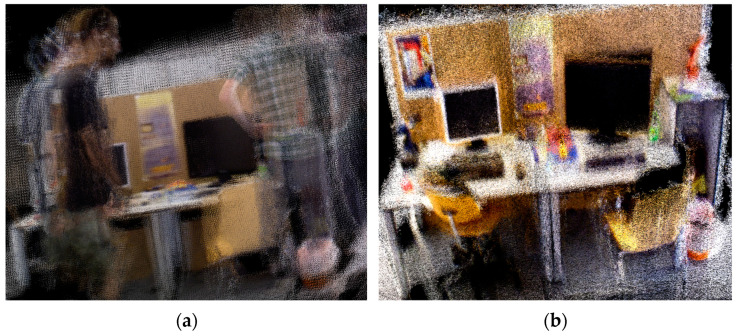
Using ORB-SLAM2 and YPR-SLAM to construct dense 3D point cloud map in dynamic scene sequence fre3_walking_xyz. (**a**) represents a dense 3D point cloud map constructed by the ORB-SLAM2 system; (**b**) represents a dense 3D point cloud map constructed by the YPR-SLAM system.

**Table 1 sensors-24-06576-t001:** This describes the results of ablation experiments on YPR-SLAM, where the ATE is measured in meters. Different methods were used in the experiments. The letter YOLO represents the YOLOv5 method, and the letter Depth-PROSAC represents the Depth-PROSAC method. The best values are in bold and ✓ indicates that the algorithm module is included.

Methods		fre3_walking_halfsphere	fre3_walking_xyz	fre3_walking_static	fr3_sitting_static
YOLO	Depth-PROSAC				
		0.3153	0.2627	0.0209	0.0067
✓		0.0219	0.0141	0.0086	0.0064
	✓	0.2929	0.2314	0.0221	0.0066
✓	✓	**0** **.0194**	**0** **.0137**	**0** **.0064**	**0** **.0057**

**Table 2 sensors-24-06576-t002:** *ATE* values for different sequences of ORB-SLAM2 system, DS-SLAM, and YPR-SLAM system. The best values are in bold.

Sequence	ORB-SLAM2		DS-SLAM		Ours		Improvements against ORB-SLAM2	
	RMSE	S.D.	RMSE	S.D.	RMSE	S.D.	RMSE	S.D.
fre3_walking_halfsphere	0.3153	0.1338	0.0314	0.0178	**0.0194**	**0.0101**	93.85	92.45
fre3_walking_xyz	0.2627	0.1217	0.0252	0.0165	**0.0137**	**0.0071**	94.78	94.17
fre3_walking_static	0.0209	0.0111	0.0086	0.0041	**0.0064**	**0.0032**	69.38	71.17
fr3_sitting_static	0.0067	0.0032	0.0064	0.0033	**0.0057**	**0.0026**	14.93	18.75

**Table 3 sensors-24-06576-t003:** RTE values of metric translation for different sequences of ORB-SLAM2 system, DS-SLAM system, and YPR-SLAM system. The best values are in bold.

Sequence	ORB-SLAM2		DS-SLAM		Ours		Improvements against ORB-SLAM2	
	RMSE	S.D.	RMSE	S.D.	RMSE	S.D.	RMSE	S.D.
fre3_walking_halfsphere	0.2223	0.1729	0.0289	0.0162	**0** **.0259**	**0** **.0140**	88.35	91.90
fre3_walking_xyz	0.2417	0.1719	0.0342	0.0234	**0** **.0191**	**0** **.0096**	92.10	94.42
fre3_walking_static	0.1618	0.1454	0.0113	0.0051	**0** **.0092**	**0** **.0047**	94.31	96.77
fr3_sitting_static	0.0095	0.0056	**0** **.0079**	**0** **.0037**	0.0080	0.0041	15.79	26.79

**Table 4 sensors-24-06576-t004:** RTE values of metric rotation for different sequences of ORB-SLAM2 system, DS-SLAM system, and YPR-SLAM system. The best values are in bold.

Sequence	ORB-SLAM2		DS-SLAM		Ours		Improvements against ORB-SLAM2	
	RMSE	S.D.	RMSE	S.D.	RMSE	S.D.	RMSE	S.D.
fre3_walking_halfsphere	4.6094	3.5249	0.8146	0.4122	**0.7086**	**0.3768**	84.63	89.31
fre3_walking_xyz	4.6623	3.3229	0.8271	0.5835	**0.5754**	**0.3706**	87.66	88.85
fre3_walking_static	3.5621	3.2356	**0.2691**	**0.1179**	0.2710	0.1252	92.40	96.13
fr3_sitting_static	0.2763	0.1441	0.2742	**0.1243**	**0.2653**	0.1306	3.98	9.37

**Table 5 sensors-24-06576-t005:** The total time consumed is in the ORB-SLAM2 system, DS-SLAM system, and YPR-SLAM system. The best values are in bold.

Sequence	ORB-SLAM2		DS-SLAM		Ours	
	Median	Mean	Median	Mean	Median	Mean
fre3_walking_halfsphere	70.73	74.92	65.38	67.58	**38.70**	**39.48**
fre3_walking_xyz	50.93	51.90	57.98	59.12	**3** **7.74**	**3** **8.94**
fr3_walking_static	54.40	55.14	64.57	66.95	**32.47**	**33.20**

## Data Availability

We The dataset used in this paper is the public TUM dataset. The download address is as follows: https://vision.in.tum.de/data/datasets/rgbd-dataset (accessed on 21 March 2024).

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
