# Peer review of "YPR-SLAM: A SLAM System Combining Object Detection and Geometric Constraints for Dynamic Scenes"

_sensors, 2024, doi:10.3390/s24206576_

Round 1

Reviewer 1 Report

Comments and Suggestions for Authors

In this manuscript, the author proposes a dynamic environment localization method that integrates neural networks to overcome the low localization accuracy of SLAM systems in dynamic environments. Research has shown that the proposed method improves the efficiency and accuracy of SLAM system localization in dynamic environments. In addition, this article also provides a solution for rapid localization of SLAM systems in dynamic environments. Here are some suggestions:

1.      In the experiment, two similar datasets were mentioned in the high dynamic scene, namely fre3_walkaing_ half and fre3_walkaing_ halfsphere. If they are two different datasets, it is recommended to supplement the experiment; If the dataset is the same, please check and modify the article for similar issues.

2.      There are formatting issues with Table 2. It is recommended to reformat the paper format.

3.      Please provide more detailed information about the introduction in lines 342-355. It is recommended to reformat and explain Figure 7 and its corresponding text.

4.      When testing the real-time performance of the method using a high dynamic scene dataset, this article only used the fre3_walking_xyz dataset. It is recommended to also test other high dynamic scene datasets mentioned earlier.

Comments on the Quality of English Language

Suggest polishing the language used in writing.

Author Response

Comment 1: In the experiment, two similar datasets were mentioned in the high dynamic scene, namely fre3_walkaing_ half and fre3_walkaing_ halfsphere. If they are two different datasets, it is recommended to supplement the experiment; If the dataset is the same, please check and modify the article for similar issues.

Answer: Thank you. fre3_walking_half and fre3_walking_halfsphere are the same sequence. Due to a writing error, the incorrect terms have been revised throughout the paper. The modified sections are highlighted in yellow.

Comment 2: There are formatting issues with Table 2. It is recommended to reformat the paper format.

Answer: Thank you. The format of Table 2 has been corrected, and the formats of other tables in the paper have also been checked and revised.

Comment 3: Please provide more detailed information about the introduction in lines 342-355. It is recommended to reformat and explain Figure 7 and its corresponding text.

Answer: Thank you.To provide a more detailed explanation of Figure 7 and better illustrate the comparison between the estimated camera trajectories and the real trajectories after running different algorithms on the dataset, additional descriptive statements have been added to clarify the content of the figure more effectively. The following words have been revised and added in paragraph 1 with yellow highlighted:

“Figure 7 shows the Absolute Trajectory Error (ATE) and Relative Pose Error (RPE) calculated using ORB-SLAM2 and YPR-SLAM. For the low dynamic scene sequence fre3_siting_static, the errors of ORB-SLAM2 and YPR-SLAM are very close to the real trajectory on the ground, indicating that both systems can provide relatively accurate trajectory estimation in this scene. However, in another low dynamic scene sequence fre3_walking_static, the Absolute Trajectory Error and Relative Pose Error of ORB-SLAM2 drift significantly, showing that the system lacks robustness in this scenario. In contrast, the YPR-SLAM system maintains high accuracy and its trajectory estimation remains stable in this scene.Further analysis of the high dynamic scene sequences fre3_walking_halfsphere and fre3_walking_xyz reveals that the trajectory estimation of ORB-SLAM2 deviates significantly from the real trajectory, with a considerable increase in error. This demonstrates the limitations of ORB-SLAM2 in handling high dynamic environments. Conversely, the Absolute Trajectory Error of the YPR-SLAM system in these high dynamic scenes is smaller and closer to the real trajectory, showing its excellent performance in such environments.From these experimental results, it is evident that the YPR-SLAM system performs superiorly in dynamic environments, especially in high dynamic scenes where it exhibits better robustness and accuracy. In contrast, ORB-SLAM2’s performance in dynamic scenes is relatively weaker, as it is more susceptible to environmental changes, leading to increased errors. The YPR-SLAM system demonstrates higher precision and consistency under different dynamic conditions, with its advantages being even more pronounced in high dynamic scenes, proving its reliability in complex dynamic environments. These results further verify the applicability of the YPR-SLAM system in dynamic environments, providing more stable and accurate trajectory estimation for real-world applications.

Comment 4: When testing the real-time performance of the method using a high dynamic scene dataset, this article only used the fre3_walking_xyz dataset. It is recommended to also test other high dynamic scene datasets mentioned earlier.

Answer: Thank you. To better test the effectiveness of this method in dynamic scenes, additional runtime experiments on dynamic scene datasets were conducted to further compare the results.

Reviewer 2 Report

Comments and Suggestions for Authors
  • The paragraph from lines 94-100 identifies key limitations but the statement is overly general and lacks depth. It should specify which systems struggle with the computational load of models and why this affects real-time performance. The issue of mistakenly filtering static points should also be expanded to explain how this reduces feature points and impacts localization accuracy. Lastly, the mention of dynamic objects in dense mapping should discuss the specific consequences, like map inconsistencies and ghosting, as seen in PR-SLAM. 
  • The use of YOLOv5 is presented without sufficient justification for why it was chosen over more recent models (YOLOv8, YOLOv9). While the authors mention they are “not as mature,” this reasoning is not sufficient scientific justification, as “maturity” is not a well-defined metric. The comparison should include technical reasons, like performance on specific datasets or computational benchmarks.
  • In line 130, “when the detected region occupies two-thirds or more of the image, the method’s effectiveness decreases”, please add references to support this numerical statements.
  • In line 201, though authors mentioned “the 2D detection box information and labels generated in the potential motion target thread are then transmitted to the subsequent threads of the ORB-SLAM2 system”, this lacks clarity on technically how this integrates with SLAM. It should specify how the dynamic regions are utilized in tracking and mapping after detection.
  • In section 3.3, while average and median times are provided, it lacks analysis of the impact of these differences on practical applications. Also,  it is unclear how the lightweight detection algorithm (YOLOv5) and Depth-PROSAC algorithm contribute specifically to the speed improvements.
  • In addition, there is no discussion of the hardware or system configuration used in the experiment, which is critical for understanding the system’s real-time performance.
  • In conclusion, there is no discussion on the broader impact of these results or potential future work.

Author Response

Comment 1: The paragraph from lines 94-100 identifies key limitations but the statement is overly general and lacks depth. It should specify which systems struggle with the computational load of models and why this affects real-time performance. The issue of mistakenly filtering static points should also be expanded to explain how this reduces feature points and impacts localization accuracy. Lastly, the mention of dynamic objects in dense mapping should discuss the specific consequences, like map inconsistencies and ghosting, as seen in PR-SLAM.

Answer: Thank you. We recognize the need to provide more detail and depth in addressing the limitations discussed in the paragraph from lines 94-100. In response, we have now explicitly identified which specific systems encounter difficulties with the computational demands of large-scale models, particularly low-power and embedded systems. We have expanded this section to explain in detail how mistakenly filtering static points leads to a reduction in the available feature points. Additionally, we have further clarified how this reduction negatively affects localization accuracy, as the system relies on these static landmarks to maintain consistent and accurate mapping. Moreover, we have elaborated on the impact of dynamic objects in dense mapping, specifically discussing issues such as map inconsistencies and ghosting effects.

Comment 2: The use of YOLOv5 is presented without sufficient justification for why it was chosen over more recent models (YOLOv8, YOLOv9). While the authors mention they are “not as mature,” this reasoning is not sufficient scientific justification, as “maturity” is not a well-defined metric. The comparison should include technical reasons, like performance on specific datasets or computational benchmarks.

Answer: Thank you. In the paper, a comparison of detection speed, detection accuracy, and network parameters between YOLOv5, YOLOv8, and YOLOv9 has been added. For edge devices, it is important not only to consider detection accuracy but also to evaluate factors such as model complexity, memory usage, and the trade-off between detection accuracy and detection speed. The following words have been revised and added in paragraph 1 with yellow highlighted:

“First, in terms of accuracy, YOLOv5 achieves an average precision of 50.5% on the COCO dataset, while YOLOv8 achieves 51.4%. Second, in terms of speed (FPS), YOLOv5 is highly optimized, reaching 98ms in the CPU version, whereas YOLOv8 takes 128.4ms. Regarding parameters, YOLOv5s has 7.2M parameters, while YOLOv8s has 11.2M. Although YOLOv8's increased number of parameters slightly improves detection accuracy, it sacrifices detection speed. Since edge devices need to consider memory and the runtime speed of SLAM algorithms, YOLOv5 is chosen [39].

Comment 3: In line 130, “when the detected region occupies two-thirds or more of the image, the method’s effectiveness decreases”, please add references to support this numerical statements.

Answer: Thank you. Due to the lack of relevant references, it has been difficult to illustrate the challenges and effectiveness of this issue. Additional references have been included to support the aforementioned points.

Comment 4: In line 201, though authors mentioned “the 2D detection box information and labels generated in the potential motion target thread are then transmitted to the subsequent threads of the ORB-SLAM2 system”, this lacks clarity on technically how this integrates with SLAM. It should specify how the dynamic regions are utilized in tracking and mapping after detection.

Answer: Thank you. The SLAM algorithm in this paper integrates the YOLOv5 inference model into the SLAM system by creating a separate thread to run the YOLOv5 algorithm, which detects each frame of the image and outputs the four corner coordinates and label information of the 2D bounding boxes for potential targets. Finally, in the tracking thread of ORB-SLAM2, the four corner coordinates and label information of the 2D bounding boxes of potential targets are obtained.The following words have been revised and added in paragraph 1 with yellow highlighted:

“Each frame of the image is detected by the YOLOv5 algorithm, which generates the coordinate points of the bounding boxes for each target in the image coordinate system, as well as the label information. The coordinates of the 2D bounding boxes for each detected target and the target's category (such as "person" or "car") are used as inputs to the YPR-SLAM system.

Comment 5: In section 3.3, while average and median times are provided, it lacks analysis of the impact of these differences on practical applications. Also,  it is unclear how the lightweight detection algorithm (YOLOv5) and Depth-PROSAC algorithm contribute specifically to the speed improvements.

Answer:Thank you. To better compare the contribution of each component to the algorithm, ablation experiments were conducted on the proposed algorithms, allowing for a more thorough evaluation of their effectiveness.

Comment 6: In addition, there is no discussion of the hardware or system configuration used in the experiment, which is critical for understanding the system’s real-time performance. In conclusion, there is no discussion on the broader impact of these results or potential future work.

Answer: Thank you. The experimental equipment can better demonstrate the simulation environment of the algorithm; therefore, an introduction to the experimental equipment has been added in Section 3. In Section 4, the conclusion has been expanded to include an outlook on future work, potential application scenarios, and directions for further improvements. The following words have been revised and added in paragraph 1 with yellow highlighted:

“The simulation experiments were conducted on a desktop computer equipped with an Intel® Xeon® Platinum 8160 CPU, 2.10GHz clock speed, 128 GB RAM, NVIDIA Quadro P4000 graphics card, and Ubuntu 18.04 operating system. The comparison algorithm was run on a MECHREVO laptop with a 12th Gen Intel® Core(TM) i5-12450H CPU, 2.00GHz clock speed, 16.0GB RAM, NVIDIA GeForce RTX 3060 graphics card, and Ubuntu 20.04 operating system. For the prior dynamic object module, the input size of the RGB-D images was 640×640. In all experiments, the feature points number N was set to 1000, and the depth threshold dt was set to 0.3 meters.

Round 2

Reviewer 2 Report

Comments and Suggestions for Authors

The revision looks good. I don't have any questions.